# Impact of Fluxapyroxad and Mefentrifluconazole on Microbial Succession and Metabolic Regulation in Rice Under Field Conditions

**DOI:** 10.3390/foods14111904

**Published:** 2025-05-27

**Authors:** Changpeng Zhang, Nan Fang, Chizhou Liang, Xiangyun Wang, Yanjie Li, Hongmei He, Xueping Zhao, Yuqin Luo, Jinhua Jiang

**Affiliations:** 1State Key Laboratory for Managing Biotic and Chemical Threats to the Quality and Safety of Agro-Products, Ministry of Agriculture and Rural Affairs Key Laboratory for Pesticide Residue Detection, Institute of Agro-Products Safety and Nutrition, Zhejiang Academy of Agricultural Sciences, Hangzhou 310021, China; cpzhang1215@zaas.ac.cn (C.Z.); fn199198@hotmail.com (N.F.); wangxiangyun2@sina.com (X.W.); yanjieli0913@outlook.com (Y.L.); hehongmei53@163.com (H.H.); zhaoxp@zaas.ac.cn (X.Z.); 2Plant Protection Quarantine and Pesticide Management Station of Zhejiang, Hangzhou 310020, China; czliang1975@163.com

**Keywords:** brown rice, residue, bacterial community, differentially expressed metabolites

## Abstract

This study systematically evaluated the residual behavior of fluxapyroxad (FXP) and mefentrifluconazole (MFZ) in rice–soil systems, alongside their soil and metabolic impacts. Analytical methods validated via linear regression (0.0001–0.05 mg/L) complied with EU guidelines, demonstrating recoveries of 71.97–114.96%, RSDs ≤ 12.12%, and effective mitigation of matrix effects (−85.08% to −76.97%) using matrix-matched calibration. Residual dissipation followed first-order kinetics, with half-lives (T_1/2_) spanning 10.83–21.00 d (FXP) and 23.10–57.76 d (MFZ). Notably, MFZ exhibited prolonged persistence in brown rice (T_1/2_ = 57.76 d), though final residues (0.031 ± 0.001 μg/g FXP; 0.011 ± 0.0003 μg/g MFZ) remained below regulatory limits (China: 1 mg/kg; CAC: 5 mg/kg). Microbial analysis revealed transient diversity loss in rhizosphere communities (Chao1 index, *p* < 0.05), recovering by 21 d, while endophytes displayed resilience linked to plant metabolites. Enrichment of degraders (e.g., *Sphingomonas*) contrasted with suppression of nitrogen-fixing *Bradyrhizobium*, indicating functional trade-offs. Metabolomic profiling identified 3512 metabolites, with 332 and 173 differentially expressed metabolites at 7 d (S) and 21 d (T), dominated by lipids, benzenoids, and phenylpropanoids. Key metabolic shifts included a 2.11-fold increase in coumarin and elevated L-aspartic acid, highlighting adaptive responses via phenylalanine and TCA cycle pathways. Correlation analyses linked stress-tolerant endophytes (*Azorhizobium*) to defense-related metabolites (e.g., coumarin), suggesting microbial modulation of plant resilience. These findings emphasize the need for integrated strategies combining residue monitoring, microbial management, and metabolic insights to mitigate agrochemical risks in sustainable agriculture.

## 1. Introduction

Rice (*Oryza sativa* L.) is one of the world’s most significant cereal crops, accounting for half of the world’s daily meals, and its steady and high yields ensure increasing global agricultural productivity [1,2]. Brown rice is an entire grain that provides the body with essential nutrients and energy, including the trace elements selenium (Se), zinc (Zn), iron (Fe), and thiamine, as well as carbohydrates, proteins, lipids, and dietary fiber [3,4,5]. Despite its nutritional value, rice production faces substantial challenges, including yield losses due to diseases like sheath blight, false smut, and rice blast, which severely compromise both yield and quality [6,7]. Scientific administration (a growth-friendly environment, pesticide application, and fertilization), suitable density planting, and fertilization are the main variables contributing to high agricultural yield [8,9,10]. According to research, crop losses without pesticides fluctuate between 40% and 70%, and when specific crops are subjected to epidemics or insect outbreaks, the losses are considerably larger [11,12]. China, for instance, reported an annual pesticide use of 2.212 kg/ha between 2016 and 2020, surpassing both Asian (1.24 kg/ha) and global (1.838 kg/ha) averages [13]. The efficient stabilization of yields from crops has been made feasible in large part by pesticide application. Nevertheless, using pesticides comes with a double-edged sword because the negative consequences (food safety, degradation, and worsening of ecosystems and surroundings) of pesticide overuse and misuse have been extensively established [14,15].

Fluxapyroxad functions by inhibiting succinate dehydrogenase in mitochondrial respiratory chain complex II and has been demonstrated to effectively suppress sheath blight and rice root-knot disease [16]. A broad-spectrum innovative triazole fungicide, mefentrifluconazole, was commercialized by BASF in 2019. Given the distinctive molecular structure of isopropanol, mefentrifluconazole has been shown to have a strong fungicidal effect and an absence of cross-resistance. In China, mefentrifluconazole has been authorized for use in ten various crops, including rice, approved in May 2024 (http://www.chinapesticide.org.cn/zgnyxxw/zwb/dataCenter, accessed on 1 March 2025), which serves to control rice false smut. In addition to rice blast (caused by *Magnaporthe oryzae*), sheath blight (*Rhizoctonia solani*) and false smut also seriously decrease the yield and quality of rice. Although fungicides such as fluxapyroxad (FXP) and mefentrifluconazole (MFZ) stabilize yields by controlling pathogens, their overuse poses risks. These include environmental degradation, ecosystem disruption, food safety concerns, and the emergence of pesticide-resistant pathogen strains due to high application doses and frequencies. The resistance of drug-resistant pathogens to benzimidazoles, triazoles, and SDHIs has been reported, threatening the disease control effect [17]. Although both fungicides are registered for use in China across multiple crops (including rice for MFZ since May 2024), their combined application—a promising strategy to delay resistance development—remains unregistered and understudied in rice systems. It is therefore critical to gather foundational data on the residue levels of fluxapyroxad and mefentrifluconazole in rice.

Statistics demonstrate that less than 10% of pesticides remain effective due to droplet evaporation, drift, plant leaf bouncing and rolling down, and other factors [18]. A huge variety of soil microbial communities that are engaged in intricate ecological and biological cycles can be found in the rhizosphere, a micro-ecological area that links plant soil and roots. In the meantime, endophytes directly contributed to the biological metabolism of the host as well as being influenced by the choice of plant hosts and environmental conditions [19]. Current research gaps hinder a comprehensive understanding of the impacts of FXP and MFZ. First, while these fungicides persist in both soil and aquatic areas [20,21], their effects on rice root-associated microbial communities—critical for nutrient cycling, plant health, and soil ecosystems—are poorly characterized [19,22]. Second, metabolomic shifts in rice under fungicide exposure, which may alter nutritional quality or sensory attributes, remain largely unexplored [18,23,24]. Such knowledge is vital to reconcile agronomic benefits with potential risks to rice quality and ecological sustainability.

This study aims to address these gaps through the following objectives: (1) Residue dynamics: to quantify FXP and MFZ residues in brown rice over 35 days post-application at 180 g.a.i./hm^2^. (2) Microbial community shifts: to investigate temporal changes in root-associated bacterial diversity, composition, and functional profiles induced by FXP and MFZ. (3) Metabolomic profiling: to identify fungicide-induced alterations in rice metabolic pathways linked to nutritional quality and defense at 7 and 21 days. By integrating residue, microbial, and metabolomic data, this work provides foundational insights for optimizing fungicide use while safeguarding rice quality and ecosystem health.

## 2. Materials and Methods

### 2.1. Materials

Mefentrifluconazole (MFZ: 98.86% purity) and fluxapyroxad (FXP: 99.89% purity) were obtained from Dr. Ehrenstorfer GmbH (Augsburg, Germany). MFZ·FXP suspension concentrate (400 g/L, SC: 200 g/L MFZ, 200 g/L FXP) was provided by BASF Plant Protection (Jiangsu) Co., Ltd. (Rudong, China). Acetonitrile (ACN: HPLC grade) and ammonium acetate (analytical grade) were purchased from Merck (Darmstadt, Germany). NaCl (analytical grade) was provided by Yida Chemical Reagent Co., Ltd. (Lanxi, China). The 2 mL purification tube with dispersed solid-phase extraction materials (octadecylsilane, pesticarb, and primary secondary amine), anhydrous magnesium sulfate (MgSO_4_), and a 0.22 μm nylon syringe filter were prepared by Tianjin Bonna-Agela Technologies Co., Ltd. (Tianjin, China). The standard stock solutions (100 mg/L) of MFZ and FXP were made in ACN and stored at −18 °C. ACN was used to dilute in succession standardized working solutions in the amounts of 0.0001, 0.0005, 0.001, 0.005, 0.01, and 0.05 mg/L of a standard stock solution. Accordingly, untreated matrix (brown rice, rice husk, rice root, and soil) extracts were thrown in each successively diluted standard solution to establish matrix-matched standard solutions at 0.0001–0.05 mg/L. All standard solutions were refrigerated at 4 °C and shielded from light.

### 2.2. Experimental Design

Rice seeds (Xiangyou 269) were directly sown in a field in Hangzhou, Zhejiang (119.33° E, 29.42° N) in July 2021. There is no prior history of MFZ and FXP, or other cognate substances, being applied at the testing site. In a field with a temperature range of 16.1–29.6 °C and humidity of 54–83%, the seedlings were raised. Two groups (a control group, without spraying MFZ · FXP suspension concentrate, and a course of treatment group, with spraying MFZ · FXP suspension concentrate) were established in an experimental field in which fungicides had not previously been utilized. The treatment group included 6 plots of 100 m^2^ each, and the control group included 6 plots of 100 m^2^ each, divided by a buffer zone.

The NY/T 788-2018 Guidelines were used for arranging the field trial [25]. At the growth (8 October) stages, an initial application of MFZ · FXP suspension concentrate (40%) at a dose of 180 g.a.i./hm^2^ was applied to the rice, and the aqueous solution was made ready for leaf spraying. We kept spraying the same amount of pesticides on the rice plants after seven days on the same plot (they were sprayed twice). The rice was hand harvested after the last spraying at an exposure interval of 1, 3, 5, 7, 14, 21, 28, and 35 days (d) for randomly collecting brown rice and rice husk by shelling the rice grain to detect dynamic residues of target pesticides in the rice system. The rice root and root soil were sampled at 7 and 21 d to investigate the responses in the root-associated bacterial community (rhizosphere and endophytic bacteria). The rice samples were taken at 7 and 21 d to explore the differential metabolites using metabolomics. The residual samples were refrigerated at −18 °C for subsequent testing. Rice samples were stored at −80 °C prior to high-throughput sequencing and metabolomic analysis.

### 2.3. MFZ and FXP Concentration Analysis

Brown rice and soil (5 g) were placed into a 50-mL centrifuge tube, and the sample was oscillated for 30 min using 5 mL of deionized water and 25 mL of ACN (rice husk and rice root: 2 g, 5 mL of deionized water and 10 mL of ACN). Then, 5 g of NaCl was progressively added, followed by a vortex extraction for 1 min and a 5 min centrifugation at 4000 rpm. A 2 mL purification tube comprising dispersive solid-phase extraction adsorbents (150 mg of MgSO_4_, 50 mg of C_18_, 50 mg of PSA, and 8 mg of PC) was filled with the resulting supernatant (1.5 mL) after extraction. The tube was then centrifuged for 5 min at 8000 rpm after being vortexed for 1 min. The supernatant was then filtered with a 0.22 μm PTFE filter for UPLC-MS/MS testing. The Appendix A offer comprehensive information on the methods and UPLC-MS/MS analyses for MFZ and FXP.

### 2.4. Illumina Sequencing and Functional Prediction for Bacterial Community

Soil genomic DNA was separated from the obtained soil via CTAB, following the manufacturer’s instructions. The DNA concentration and quality were then assessed using a UV-based spectrophotometer and 2% agarose gel electrophoresis, respectively. Sequencing library generation was performed using the primer pair 341F (5′-CCTACGGGNGGCWGCAG-3′) and 805R (5′-GACTACHVGGGTATCTAATCC-3′), the V3-V4 hypervariable region of the bacterial 16S rRNA gene was amplified, and then the PCR products were purified and quantified using AMPure XT beads (Beckman Coulter Genomics, Danvers, MA, USA) and Qubit (Invitrogen, Waltham, MA, USA), respectively. Purified PCR products from each DNA sample were merged and sequenced using the Illumina NovaSeq platform (Kapa Biosciences, Woburn, MA, USA) at LC-Bio Technology (Hangzhou, China). Amplicon sequence variants (ASVs) were identified using DADA2, which clusters sequences at 100% similarity. The relative abundance of each sample was used by the SILVA (version 138) classifier and the NT-16S database to normalize feature abundance and annotate at several taxonomic levels (phylum, class, family, and genus), respectively. We utilized the most recent release of the PICRUSt2 pipeline (https://github.com/picrust/picrust2, accessed on 23 July 2023) to ascertain the latent metabolic functions of the bacterial community based on the KEGG database that uses the characteristic sequence [26]. STAMP software (version 3.2) was employed to statistically determine the variation in predicted functions between the groups being tested.

### 2.5. Extraction of Metabolites and Metabolomics Analysis

The extraction mixture was stored overnight at −20 °C after being combined with 100 mg of rice sample and 1 mL of precooled fifty percent methanol, swirled for 1 min, and finally placed at room temperature for 10 min. The supernatants were then put onto clean 96-well plates after 20 min of centrifugation at 4000× *g*. Before the LC-MS analysis, the samples were kept at −80 °C. Furthermore, 10 μL of each extraction mixture was combined to generate pooled QC samples.

A liquid chromatography system (Vanquish Flex UHPLC, Thermo Fisher Scientific, San Jose, CA, USA) with an ACQUITY UPLC T_3_ column (100 mm × 2.1 mm, 1.8 μm, Waters, Milford, MA, USA) equipped with a quadrupole mass spectrometer (Q-Exactive, Thermo Fisher Scientific, San Jose, CA, USA) was used to evaluate the amounts of various metabolites. Detailed information about the analysis software involved in the metabolome is as follows: Compound Discoverer 3.1.0 (Thermo Fisher Scientific, San Jose, CA, USA) for raw spectra, determination of charge mass ratio, and compound identification. Metabolite identification was combined with the in-house metabolite secondary mass spectrometry library, mzCloud, mzVault, Mass list, OTCML, ChemSpider (KEGG, LipidMaps, PlantCyc, Planta Piloto de Quimica), etc. Finally, information such as the molecular weight, retention time, peak area, and metabolite identification results of each ion was obtained. Annotation: HMDB and KEGG databases. Structural prediction: SIRIUS (v5.5.6).

## 3. Results and Discussion

### 3.1. Method Validation of MFZ and FXP in Samples

According to An et al. [27], ACN was chosen as an extractive solvent. Through the use of a linear regression approach, the method’s linearity (0.0001–0.05 mg/L) was validated (Appendix A). Two fungicides had appropriate coefficients of determination (R^2^ ≥ 0.9914 in all matrices). The lowest spiking level of an analytical procedure is identified as the limit of quantification (LOQ). Therefore, the LOQ for the technique was 0.01 mg/kg. The precision of target pesticide detection can be influenced by matrix effects. Thus, a newly developed approach should assess the matrix effect and try to eliminate the effect of ionization on the target analyte during co-elution of the matrix constituent. Matrix effects in rice husks of this research varied from −85.08% (FXP) to −76.97% (MFZ) (Appendix A), demonstrating a strong suppression of ionization in rice husks compared with other matrices. Thus, matrix-matched standard curves were used in this work’s quantitative study of MFZ and FXP to do away with the matrix effect’s interference.

Relative standard deviations (RSDs) and recoveries were used to assess the method’s accuracy and precision (Table 1). All matrix mean recoveries (n = 24) ranged from 71.97% to 114.96% at three spiking levels (0.01, 0.1, and 5.0 mg/kg) with five repetitions, and the RSDs of MFZ and FXP ranged from 1.23% to 12.12%. The method complies with the European Union guidelines (SANTE/12682/2019) for pesticide residue analysis [28].

### 3.2. The Residues and Dissipation of FXP and MFZ in Rice–Soil Systems

As shown in Figure 1A, the amounts of FXP and MFZ in rice husk fluctuated between 7.293 ± 0.398 (1 d) and 0.879 ± 0.030 μg/g (28 d), 4.975 ± 0.090 (1 d) and 0.593 ± 0.038 μg/g (28 d), respectively. Residues of FXP and MFZ in rice husk were reduced by 79.33% and 67.85% at 3 d, respectively. Then, the residues of FXP and MFZ increased to reach 79.57% and 79.36%, respectively, by 14 d (compared with 7 d); the transient increase in FXP and MFZ residues at 14 d may be attributed to environmental redistribution (for example, rainfall-mediated transport). The dissipation of FXP and MFZ conformed to first-order kinetics well in rice husk, with R^2^ values of 0.8666~0.9302. The calculated T_1/2_ values of FXP and MFZ were 10.83 and 23.10 d in rice husk, respectively. Rice husk residues may transfer to bran during milling. Despite FXP’s shorter T_1/2_, its higher initial residues (7.293 μg/g at 1 d) necessitate stringent post-harvest handling to avoid bran contamination. For brown rice (Figure 1B), the beginning residues (the highest value) at 1 d were 0.049 ± 0.001 (FXP) and 0.041 ± 0.0005 mg/kg (MFZ), and the greatest reductions in residue of FXP (61.91%) and MFZ (54.71%) were at 3 d (relative to 1 d). The concentrations of FXP and MFZ in brown rice gradually rose from 5 d to 14 d. Then, the residue in brown rice of FXP and MFZ gradually declined over the course of 14 d to 35 d, yet only the dissipation of MFZ matched the first-order kinetic equation, with a matching T_1/2_ value of 57.76 d (Appendix A). Chronic exposure to triazole derivatives like MFZ is linked to endocrine disruption. The T_1/2_-driven persistence underscores the need for residue monitoring in brown rice.

As for rice root (Figure 1C), the concentrations of FXP and MFZ peaked at 0.061 ± 0.005 and 0.062 ± 0.009 μg/g at 35 d, which were 13.63 and 9.38 times higher than at 1 d, respectively. FXP and MFZ contents in roots eventually became consistent. FXP and MFZ peaked in soil (Figure 1D) at concentrations of 0.084 ± 0.011 and 0.065 ± 0.003 μg/g (1 d), respectively. FXP in rice soil was removed at a rate of 6.20% at 3 d, which was less than the removal rate at 35 d (7.88%). The amount of MFZ decreased by 24.91% and 16.18%, respectively, at 21 and 35 d in soil samples, which was much lower than the reduced amount at 3 d (31.75%). This pattern indicated that the degradation of MFZ by soil microorganisms gradually increased after the initial stage. The dissipation of FXP and MFZ in soil followed first-order kinetics, with R^2^ of 0.9033 and 0.8666, and the T_1/2_ values were 21.00 d and 25.67 d (Appendix A). The longer T_1/2_ of MFZ in soil (compared to FXP) suggests higher persistence, potentially prolonging its interaction with soil microbiota. Also, the longer T_1/2_ of MFZ across all matrices highlights its higher environmental and food safety liability compared to FXP.

The residues of FXP and MFZ on rice root, rice husk, and brown rice are more regular than those detected in soil, which may be attributed to the soil’s microbiota as well as its dilution and diffusion properties. According to an increasing order, the average residues of FXP and MFZ were on rice husk ≫ soil > rice root > brown rice, which was dependent on foliar spraying of FXP and MFZ. The half-life (T_1/2_) of FXP and MFZ varied significantly across all matrices (soil, rice husk, and brown rice), reflecting distinct environmental and agronomic risks. Notably, MFZ exhibited prolonged persistence in brown rice (T_1/2_ = 57.76 d), raising concerns over chronic dietary exposure linked to endocrine disruption. The moderate hydrophobicity of FXP (Log Kow: 3.13) and MFZ (Log Kow: 3.4) and soil T_1/2_ values indicate the potential for gradual leaching into groundwater, especially in regions with high rainfall or sandy soils. Final residues of FXP and MFZ in brown rice (0.031 ± 0.001 μg/g and 0.011 ± 0.0003 μg/g at 35 d) were below MRLs set by China (1 mg/kg) and the Codex Alimentarius Commission (CAC, 5 mg/kg). The results indicate that FXP and MFZ in rice do not pose a dietary risk.

### 3.3. Effects of FXP and MFZ Residues on Bacterial Diversity and Composition

The whole soil–root sample coverage values achieved 0.99 (Appendix A), indicating that the samples were sequenced at a sufficient sequencing depth. The high-quality sequences, with sequence counts ranging from 65,258 to 77,884 (69,574 to 84,282 in endophytes) per sample in the rhizosphere, were gathered from 24 samples and clustered into 1122–3511 and 450–2032 ASVs in the rhizosphere (soil) and endophytes (root), respectively. Alpha diversity (Chao1 index: *p* < 0.05, Appendix A) temporarily decreased at 7 days post-application (S treatment) but rebounded significantly by 21 days (T treatment), indicating transient stress followed by microbial adaptation. Endophytic communities (Appendix A) exhibited greater resilience, with diversity fluctuations less pronounced than in the rhizosphere, likely due to buffering by plant-derived metabolites. PCoA analysis (Appendix A) revealed distinct clustering of microbial communities under FXP and MFZ residues, except for endophytes at 7 days, suggesting stronger initial perturbation in the rhizosphere. These results demonstrated that low endophyte bacterial diversity (e.g., Chao1 index) compared to the rhizosphere at 21 d was caused by high susceptibility. Conversely, the rhizosphere was characterized by a more abundant and stable microbial community structure, and plant root exudates had the ability to actively influence the diversity of root-associated microbes under FXP and MFZ residues through the considerable enrichment of different particular bacterial taxa [29].

Illumina high-throughput sequencing was used to explore the microbial community structures in soil–root samples. In total, 24 soil samples (rhizosphere) contained 77 bacterial phyla, 195 classes, 426 orders, 665 families, and 1217 genera. Proteobacteria, Acidobacteriota, and Chloroflexi were the most prevalent bacteria at the phylum level in all soil samples, with proportions of 25.74–35.71%, 17.16–22.56%, and 4.51–6.98%, respectively (Appendix A). The three predominant bacterial genera (relative abundance >1.10% in 24 soil samples) were *SC-I-84_unclassified*, *Subgroup_7_unclassified*, and *Acidobacteriales_unclassified* (Figure 2A). As for all root samples, 24 samples (endophytes) contained 61 bacterial phyla, 179 classes, 371 orders, 622 families, and 1248 genera. Proteobacteria (60.60–96.87%), Bacteroidota (0.28–20.21%), and Actinobacteriota (0.50–6.97%) were the three prominent phyla that made up the majority of the 61 bacterial phyla, and they accounted for 66.93~97.81% of all the bacteria in the root (Appendix A). The top three bacterial genera were *Rhodocyclaceae_unclassified*, *Caldimonas*, and *Bradyrhizobium* (Figure 2B).

The results showed that there were more bacterial genera that were down-regulated than those that were up-regulated; the reverse pattern was observed in the S treatment, which suggested that the pressure caused by FXP and MFZ residues had enriched a number of distinctive bacterial genera in the rhizosphere and endophytes to increase adaptability. For example, ten shared differential genera (*Dongia*, *Massilia*, *Variovorax*, *Methylomicrobium*, *Bryobacter*, *Thiobacillus*, *Lysobacter*, *Luedemannella*, *Terrimonas*, and *OM27_clade*) were identified in the rhizosphere (Appendix A). Enrichment of degraders (*Sphingomonas*) suggests accelerated pesticide breakdown, reducing environmental persistence. Up-regulation of *Lysobacter* and *Massilia* may bolster disease suppression, indirectly benefiting soil–plant health. And some genera (e.g., *Massilia*, *Methylomicrobium*, *Variovorax*, *Lysobacter*, *Terrimonas*, and *OM27_clade*) exhibited continuous up-regulation or down-regulation during S and T treatments. *Lysobacter* (up-regulated) produces lytic enzymes to suppress plant pathogens. *Massilia* (up-regulated) is known for xenobiotic degradation and plant growth promotion. In addition, the differential genera *RBG-16-49-21*, *Labrys*, *Luteibacter*, *Methylosinus*, *Pyrobaculum*, *Legionella*, *Azorhizobium*, *Phenylobacterium*, *Mycobacterium*, and *Dactylosporangium* were simultaneously discovered in endophytes in S and T treatments (Appendix A). Especially, in the S or T treatment, the relative abundances of three genera (*Labrys*, *Azorhizobium*, and *Luteibacter*) were continuously up-regulated. However, the genus *Bradyrhizobium*, a nitrogen-fixing genus, whose decline may temporarily impair soil fertility, was down-regulated.

The long-lasting selective pressure from the FXP and MFZ residues aided in the colonization of some particular functional microorganisms in the rhizosphere [30]. *Sphingomonas*, a genus known to have a pesticide biodegradation capacity, showed a dynamic adjustment in relative abundance (D7: 2.24% → Control: 1.05%, D21: 1.47% → Control: 0.94%), illustrating the adaptive response of soil microbiota, and the same trend was observed for root endophytes (D7: 2.44% → Control: 0.94%, D21: 2.24% → Control: 2.03%). While FXP and MFZ enriched taxa with degradative capacities, their persistence (for example, MFZ T_1/2_ = 25.67 d in soil) may impose chronic selective pressure, favoring stress-adapted but functionally narrow communities. The dynamic restructuring of microbial communities under FXP/MFZ residue reflects a balance between functional adaptation and ecological cost. Targeted enrichment of degraders and symbionts may mitigate pesticide residues and support plant health, but long-term monitoring is essential to prevent keystone function loss. Optimizing application timing (for example, aligning with microbial recovery phases) could harness these adaptive responses while safeguarding soil health. This regulation of plant–microbe interactions provides new proof for the ecological appropriateness of the compounded formulation.

### 3.4. Functional Prediction of Rhizospheric Bacterial Communities

All samples’ amplicon sequences (S and T) were evaluated, and PICRUSt2 was used to determine probable functions. A total of 425 distinct functions were calculated to change for the S treatment, of which 50 were significant alterations, and differential functioning accounts for 11.76%. Aerobic respiration I (cytochrome c) had the highest percentage (more than 0.01%) of significant descriptions (Appendix A), and was remarkably higher than other anaerobic respiration (purine nucleobase degradation I). Particularly, the S treatment showed a marked reduction in the abundance of aerobic respiration I (cytochrome c), which indicated that aerobic microorganisms were the dominant microbiota in soil and their activity was significantly reduced after spraying FXP and MFZ at 7 d. Of the 421 distinct functions, 192 in the T treatment had abundances that were considerably changed, and differential functionality comprised 45.61% of the total. The top three (>0.004%) descriptions of significant changes were L-histidine biosynthesis (up), 6-hydroxymethyl-dihydropterin diphosphate biosynthesis I (down), and 6-hydroxymethyl-dihydropterin diphosphate biosynthesis III (Chlamydia) (down). The number of considerably decreased descriptions is significantly higher than the number of significantly increased descriptions (Appendix A), indicating a decline in the functional activity (TCA cycle VII (acetate producers), taxadiene biosynthesis (engineered), etc.) of the relevant microbiota. The proportion of different functions showed the greatest variation from the PICRUSt2 results, implying that the community function has experienced considerable alterations.

### 3.5. Effects of FXP and MFZ on Metabolic Profiles in Rice Samples

#### 3.5.1. Metabolomic Profiling

The widely targeted metabolomic investigation with UPLC-MS/MS was conducted on rice samples to gain insight into the potential impacts of the MFZ and FXP residues on the physiological and metabolic processes in rice tissues. We identified 3512 metabolites (2348 ESI+, 1164 ESI−) with significant separation in PLS-DA (Q^2^ > 0.93, R^2^ > 0.98: Appendix A), and notable separations were observed in both the S (Figure 3A) and T treatments (Figure 3B). The S and T treatments (Figure 3C) were distinguished using 332 (143 up-regulated and 189 down-regulated) and 173 (77 up-regulated and 96 down-regulated) differential expressions of metabolites (DEMs), respectively. The 332 DEMs in the S treatment (7 days) and 173 DEMs in the T treatment (21 days) were dominated by lipids, benzenoids, and phenylpropanoids (Appendix A).

The top three DEMs in LF7D were lipids and lipid-like molecules (Glycerophospholipids, fatty acyls, and steroids and steroid derivatives, respectively), benzenoids (Benzene and substituted derivatives, phenols, and naphthalenes, etc.), and organoheterocyclic compounds. Prostaglandin f3 is a bioactive lipid mediator with anti-cancer and anti-inflammatory effects. In contrast, both prostaglandin j2 and prostaglandin f3, which are subordinate to the lipid class, were down-regulated. In addition, salicylic acid and adenine, which have a variety of positive effects on plant growth, also showed significant down-regulation. Meanwhile, the top three DEMs in LF21D were lipids and lipid-like molecules, benzenoids, and organic oxygen compounds. Flavonoids have antiviral, anti-free radical, antioxidant, lipid peroxidation reduction, anti-inflammatory, anti-aging, anti-cancer, and cardiovascular disease prevention effects. Ten flavonoids were detected in all samples, namely neodiosmin, diosmetin, leptosin, rhoifolin, luteolin, narirutin 4′-glucoside, luteolin, hesperidin, naringenin, and subulin, while a significant decrease was only detected in the content of hesperidin (LF21D) in the DEMs.

Volcano plots of DEMs are shown in Appendix A. These DEMs were clustered by hierarchical clustering analysis (Appendix A). Detailed metabolite classifications and pathway enrichments were added to Appendix A. Venn diagrams revealed that there were 26 shared DEMs across days 7 and 21 (Figure 3D) linked to rice’s adaptive response. Day 21’s down-regulated DEMs decreased by 96, but the percentage of down-regulated DEMs increased, suggesting that the difference between the LF group and CK group shrank on D21. Based on these findings, it can be tentatively inferred that the DEMs were significantly reduced on D21, indicating that the rice received fewer pesticide effects.

#### 3.5.2. Impact on Rice Aroma-Related Phenolic Compounds

Rice is made up of complex chemicals, and the sum of many different volatile molecules determines the quality of its aroma. Minor variations in rice fragrance quality significantly influence consumer preferences. The scent of rice depends on specific volatile chemicals in rice. In this study, 296 volatile chemical variables were screened as a reference to determine how FXP and MFZ application will affect rice aroma [31,32]. The relative contents of the three detected substances (benzaldehyde, indole, and palmitic acid) did not show significant changes. However, the contents of the three phenolic substances (gingerdione, coniferylaldehyde, and 2-tert-butyl-4-methoxyphenol) changed significantly, with a consistent and significant increase in the content of 2-tert-butyl-4-methoxyphenol in S and T treatments. Phenolics are crucial markers for managing rice’s aroma, but their proportions also have an impact.

#### 3.5.3. Metabolic Pathway Regulation

KEGG pathway enrichment analysis revealed that DEMs were associated with diverse metabolic pathways under LF7D and LF21D treatments. The comparison group of LF7D and CK7D (Figure 4A) revealed that 138 out of 332 DEMs could be annotated to 70 existing pathways in the database, with significant enrichment in metabolic pathways, biosynthesis of secondary metabolites, and biosynthesis of phenylpropanoids. While the microbial metabolism in diverse environments was significantly down-regulated, metabolic pathways contained the highest number of DEMs (19 metabolites), representing the most significantly altered pathway and highlighting substantial metabolite differences in the S group. In the T comparative group (Figure 4B), only 23 out of 173 differential metabolites could be annotated to 15 existing pathways in the database, of which metabolic pathways, microbial metabolism in diverse environments, and caprolactam degradation were significantly enriched, while the biosynthesis of secondary metabolites was significantly down-regulated, but no pathway contained more than five DEMs, and the pathway with the most differential metabolites was the metabolic pathway (with a total of five metabolites), but the most notable pathway was caprolactam degradation (*p* value of 0.0012).

The observed increase in fumaric acid levels likely accelerated the tricarboxylic acid (TCA) cycle, potentially explaining the elevated L-aspartic acid content in S-treated rice samples (Figure 4C). L-aspartic acid helps plants grow and develop normally even in harsh conditions by improving their physiological processes and providing the building blocks for protein synthesis. The substantial increase in carbon metabolism in the S treatment is thought to have contributed to the large decrease in the phenylalanine metabolism pathway. Plant development is intimately linked to the phenylalanine metabolic pathway, a significant secondary metabolic process in plants. Coumarin, a key component of phenylalanine metabolism, showed a 2.11-fold increase in the S-treated group, likely resulting from the down-regulation of alternative secondary metabolic routes, including salicylic acid and caffeic acid synthesis pathways, suggesting that rice prioritizes carbon metabolism and phenylalanine-derived pathways to counteract pesticide-induced stress, potentially through resource reallocation to secondary metabolite synthesis (e.g., coumarin) as a detoxification strategy.

### 3.6. Correlation Tests Between Bacterial Endophytes and Metabolites of Rice

Correlation tests were used to analyze the relationship between the top 10 endophytes and the differential expression of the top 15 metabolites more deeply. It was clear that 10 bacterial genera were significantly correlated with 15 differential metabolites (Appendix A). Caldimonas was significantly negatively correlated with five metabolites (bicyclo prostaglandin e2, prostaglandin k2, cytidine, cholinesulfate, and glycerophosphocholine) but positively correlated with one metabolite, 1-phenylpropane-1_2-dione (*p* < 0.05). Massilia was significantly and negatively correlated with three metabolites (cytidine, cholinesulfate, and glycerophosphocholine) and positively correlated with 1-phenylpropane-1_2-dione. We highlight *ANPR* and *Azorhizobium* as pivotal genera showing strong correlations with phenylpropanoids (e.g., coumarin) and L-aspartic acid (Appendix A). The negative correlation between ANPR and cytidine suggests the microbial modulation of plant defense pathways. Enriched microbial taxa (such as *Azorhizobium*) can be linked to metabolic pathways (such as phenylpropanoid suppression and TCA cycle acceleration) under pesticide stress.

Similarly, as shown in Appendix A, the association analysis revealed the heat map connecting endophytes to metabolites, establishing links within compounds and at biological levels. Only *Rhodocyclaceae_unclassified* was significantly and positively correlated with the top 10 metabolites, while the remaining nine bacterial genera showed negative correlations with differential metabolites. For example, genus *ANPR* showed a significant negative correlation with 2-oxoglutaramate, 3-vinyl-2-pyrrolidinone, dl-stachydrine, 2,3-pentadienedioic acid, xestoaminol c, Metabolite 2, and 15(r)-lipoxin a4. Most notably, genera *Aquicella*, *Rhizobium*, and *Herminiimonas* were significantly and negatively correlated with all top 15 metabolites. *ANPR* and *Azorhizobium*, the shared genera across S and T treatments, showed significant enrichment, suggesting that their metabolic capacities were activated to counteract environmental stress, and it can also be deduced that endophytes shape the process of biosynthesis of some metabolites. Enriched stress-tolerant endophytes (e.g., *Azorhizobium*) could be harnessed as bioinoculants to enhance resilience, reducing pesticide dependency [33].

## 4. Conclusions

This study systematically validated an analytical method for quantifying FXP and MFZ residues, achieving compliance with EU guidelines. Dissipation dynamics of both fungicides revealed matrix-dependent persistence, with MFZ exhibiting prolonged half-lives in brown rice (T_1/2_ = 57.76 d) and soil (T_1/2_ = 25.67 d), highlighting its potential environmental and dietary risks. While final residues in brown rice (0.031 μg/g FXP; 0.011 μg/g MFZ) fell below regulatory limits (China: 1 mg/kg; CAC: 5 mg/kg), transient residue spikes in rice husk (14 d) and soil microbe-mediated degradation patterns underscore the need for post-application monitoring. Microbial analysis demonstrated rhizosphere resilience, with alpha diversity recovering by 21 d post-treatment, whereas endophytes exhibited stability linked to plant metabolite buffering. Root exudates enriched stress-adapted taxa, suggesting microbial modulation of pesticide impacts. These findings advocate for optimized application timing aligned with microbial recovery phases and targeted bioaugmentation strategies to mitigate leaching risks. Future studies should prioritize field trials to validate long-term soil health impacts under real-world agricultural practice conditions.

## Figures and Tables

**Figure 1 foods-14-01904-f001:**
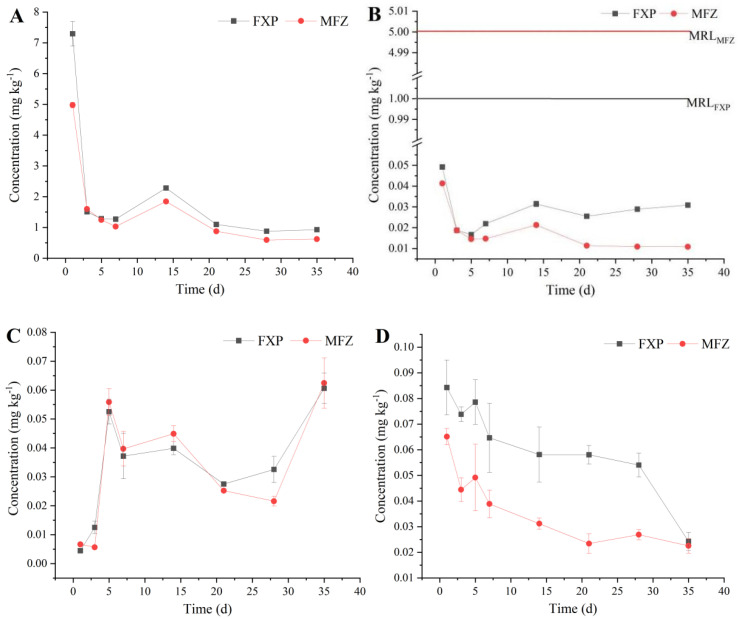
The concentrations of MFZ and FXP over time in rice husk (**A**), brown rice (**B**), rice root (**C**), and soil (**D**), respectively.

**Figure 2 foods-14-01904-f002:**
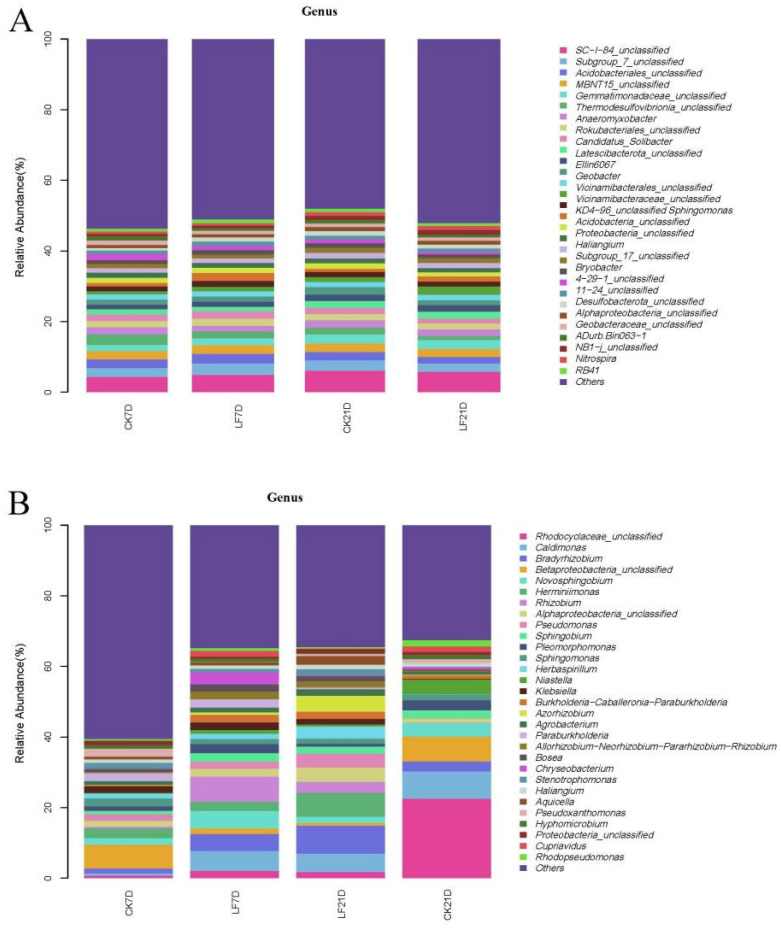
The abundance changes in the microbial consortium in CK7D, LF7D, CK21D, and LF21D at the genus level ((**A**): rhizosphere, (**B**): endosphere).

**Figure 3 foods-14-01904-f003:**
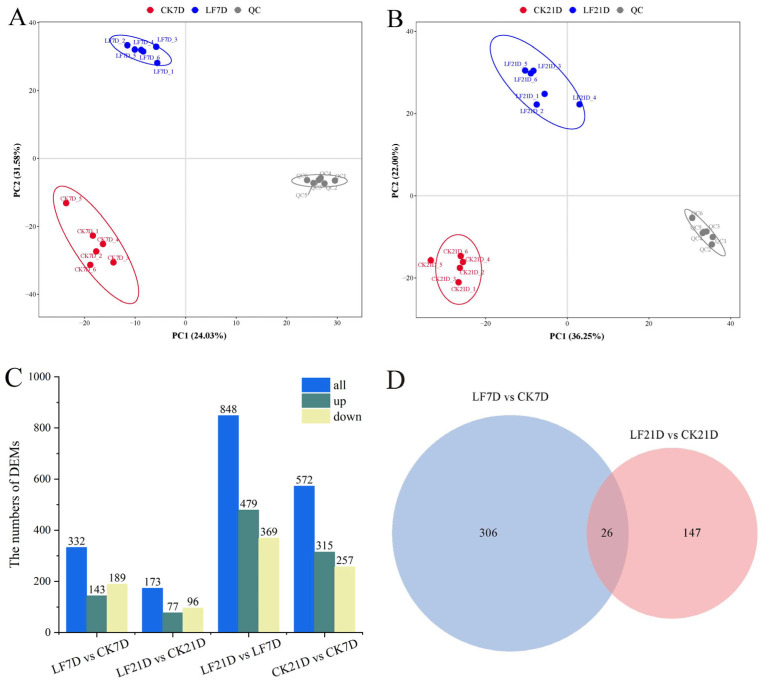
Changes in the metabolic profile of rice samples after spraying fluxapyroxad and mefentrifluconazole at 7 d and 21 d (n = 6). PLS-DA of S (**A**) and T (**B**) treatment rice metabolites in positive and negative ion mode. (**C**) The numbers of DEMs in rice samples after spraying fluxapyroxad and mefentrifluconazole. (**D**) The Venn diagram shows the overlap of the DEMs among the S and T treatments.

**Figure 4 foods-14-01904-f004:**
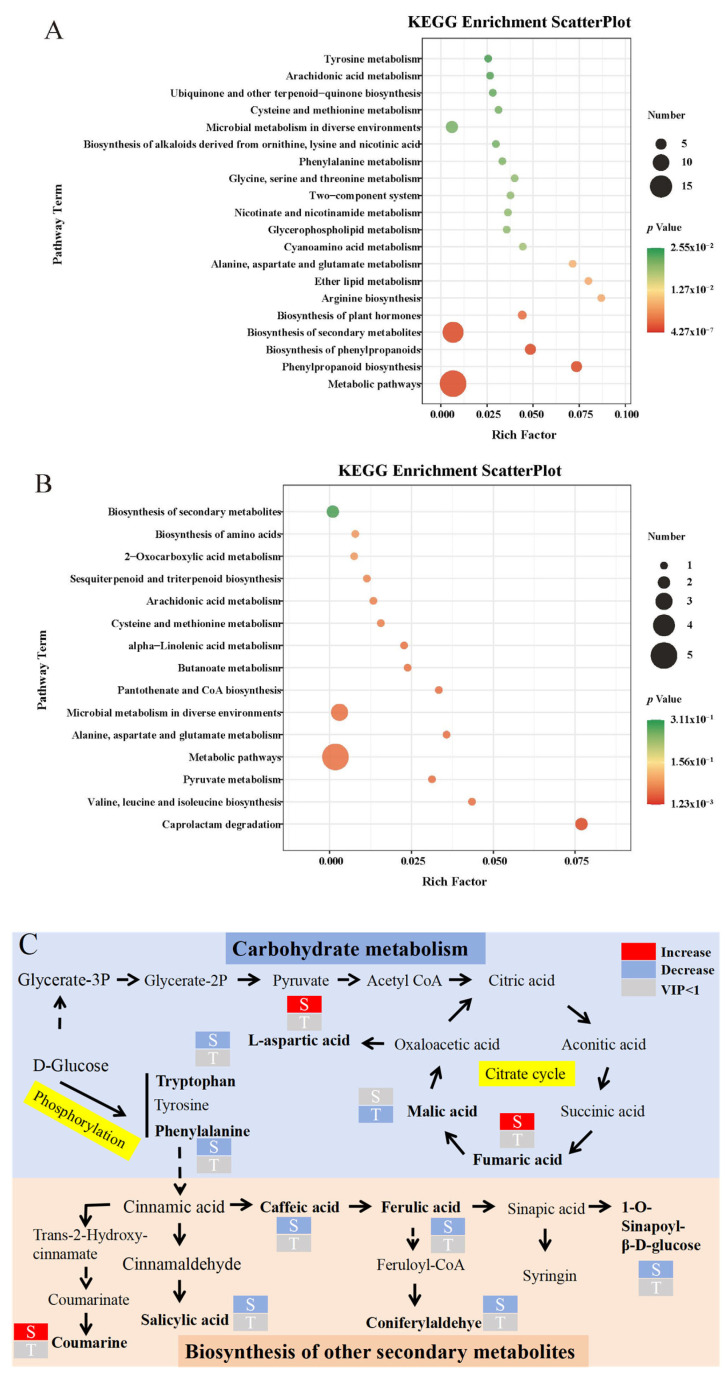
Significantly changed pathways based on enrichment analyses in S (**A**) and T (**B**) treatments. Metabolic pathways affected (**C**) in FXP with MFZ-sprayed and FXP with MFZ-free rice at 7 (marked as S) and 21 (marked as T) days.

**Table 1 foods-14-01904-t001:** Recoveries and relative standard deviations (RSDs) for FXP and MFZ in four spiked matrices at three spiked levels (n = 5).

Matrices	Spiked Levels (mg/kg)	FXP		MFZ	
		Recoveries	RSDs (%)	Recoveries	RSDs (%)
brown rice	0.01	105.79	1.82	81.12	5.17
	0.1	71.97	1.23	78.35	2.46
	5.0	109.44	4.00	86.33	5.86
rice husk	0.01	114.96	1.23	113.61	3.58
	0.1	92.85	8.76	97.92	2.38
	5.0	97.06	5.88	89.17	10.59
soil	0.01	85.81	12.12	77.56	8.01
	0.1	73.54	2.30	76.32	3.08
	5.0	106.30	3.74	100.53	3.67
rice root	0.01	77.81	6.21	82.69	6.83
	0.1	79.10	8.29	74.92	6.26
	5.0	83.59	10.08	108.60	6.68

## Data Availability

The original contributions presented in the study are included in the article/Appendix A. Further inquiries can be directed to the corresponding authors.

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
