# Peer review of "Impact of Fluxapyroxad and Mefentrifluconazole on Microbial Succession and Metabolic Regulation in Rice Under Field Conditions"

_foods, 2025, doi:10.3390/foods14111904_

Round 1
Reviewer 1 Report
Comments and Suggestions for Authors
The manuscript addressed field assessment of dual fungicide application in terms of residue dissipation, mcrobial community shifts, and rice metabolic responses. The manuscript is well-presented overall. However, the following points should be considered for further improvement:
1- The title should be revised and specific such as "Impact of Fluxapyroxad and Mefentrifluconazole on Microbial Succession and Metabolic Regulation in Rice Under Field Conditions"
2- Introduction should be orgnized to ensure flow of ideas like
-
-
Importance of rice as a crop and challenges in its production.
-
Role of fungicides and risks of overuse.
-
Specifics on fluxapyroxad and mefentrifluconazole.
-
Knowledge gaps: microbial and metabolomic effects.
-
Study objectives clearly listed in bullet or numbered format.
-
- Explain the temporary increase in residues after initial decline. Is it due to analytical variability, reabsorption, or environmental redistribution?
- Elaborate on the implication of T₁/₂ values for FXP and MFZ in different matrices. This helps link dissipation behavior to environmental and food safety concerns
- Interpret how FXP and MFZ affect microbial community structure. For example, which bacterial taxa are enriched or suppressed, and what might that mean for soil health?
- Elaborate more on the potential consequences for plant growth, nutrient cycling, or bioremediation due to microbial shift.
- The manuscript should be thoroughly revised by an English language expert to ensure clarity, grammatical accuracy, and professional tone
- Ensure that all scientific names are italicized
The English could be improved to more clearly express the research.
Author Response
Response to Reviewer 1:
The manuscript addressed field assessment of dual fungicide application in terms of residue dissipation, microbial community shifts, and rice metabolic responses. The manuscript is well-presented overall. However, the following points should be considered for further improvement:
Question 1: The title should be revised and specific such as "Impact of Fluxapyroxad and Mefentrifluconazole on Microbial Succession and Metabolic Regulation in Rice Under Field Conditions".
[Reply]: The title has been revised to “Impact of Fluxapyroxad and Mefentrifluconazole on Microbial Succession and Metabolic Regulation in Rice Under Field Conditions”.
Question 2: 2- Introduction should be orgnized to ensure flow of ideas like: a.Importance of rice as a crop and challenges in its production. b.Role of fungicides and risks of overuse. c.Specifics on fluxapyroxad and mefentrifluconazole. d.Knowledge gaps: microbial and metabolomic effects. e.Study objectives clearly listed in bullet or numbered format.
[Reply]: Thanks for your kind advice. We have revised it in Introduction part.
Question 3: Explain the temporary increase in residues after initial decline. Is it due to analytical variability, reabsorption, or environmental redistribution?
[Reply]: Thanks for your advice. This section has been rewritten in lines 214-216 underlined in red: the transient increase in FXP and MFZ residues at 14 d may be attributed to environmental redistribution (for example, rainfall-mediated transport).
Question 4: Elaborate on the implication of T1/2 values for FXP and MFZ in different matrices. This helps link dissipation behavior to environmental and food safety concerns.
[Reply]: Thanks for your advice. We have revised it in ‘3.3. The residues and dissipation of FXP and MFZ in rice-soil systems’.
Question 5: Interpret how FXP and MFZ affect microbial community structure. For example, which bacterial taxa are enriched or suppressed, and what might that mean for soil health?
[Reply]: Thanks for your advice.The application of fluxapyroxad (FXP) and mefentrifluconazole (MFZ) significantly altered the structure and diversity of soil-root-associated microbial communities, with distinct responses observed in the rhizosphere and endophytic compartments. (1) Short-term suppression, Long-Term Resilience: ① Rhizosphere: Alpha diversity (Chao1 index) temporarily decreased at 7 days post-application (S treatment) but rebounded significantly by 21 days (T treatment), indicating transient stress followed by microbial adaptation. ② Endophytes: Endophytic communities exhibited greater resilience, with diversity fluctuations less pronounced than in the rhizosphere, likely due to buffering by plant-derived metabolites. (2) Taxa-specific responses. Enriched taxa (Adaptive Groups): ① Rhizosphere: Massilia (upregulated): Known for xenobiotic degradation and plant growth promotion. Lysobacter (upregulated): Produces lytic enzymes to suppress plant pathogens. Sphingomonas (dynamic adjustment): Degrades aromatic pesticides and enhances pollutant detoxification.Thiobacillus (upregulated): Involved in sulfur/nitrogen cycling, potentially mitigating pesticide-induced redox imbalances. ② Endophytes: Azorhizobium and Luteibacter (continuously upregulated): Enhance nitrogen fixation and stress tolerance. Novosphingobium (enriched): Degrades complex organics (e.g., polycyclic aromatics) and supports plant stress adaptation; Suppressed taxa (Sensitive Groups): Rhizosphere: RBG-16-49-21 (significantly reduced): A poorly characterized oligotrophic group potentially critical for organic matter turnover. Bradyrhizobium (downregulated): A nitrogen-fixing genus, whose decline may temporarily impair soil fertility. (3) Implications for soil health. ① Positive effects. Enrichment of degraders (Sphingomonas) suggests accelerated pesticide breakdown, reducing environmental persistence. Upregulation of Lysobacter and Massilia may bolster disease suppression, indirectly benefiting soil-plant health. Functional Redundancy: Convergent restoration of community structure by 21 days indicates resilience, maintaining critical processes like nutrient cycling. ② Negative risks. Loss of Keystone Taxa: Suppression of oligotrophs (RBG-16-49-21) and nitrogen fixers (Bradyrhizobium) could disrupt organic matter decomposition and nitrogen availability. Soil function influence: While FXP and MFZ enriched taxa with degradative capacities, their persistence (e.g., MFZ T1/2 = 25.67 d in soil) may impose chronic selective pressure, favoring stress-adapted but functionally narrow communities.
Question 6: Elaborate more on the potential consequences for plant growth, nutrient cycling, or bioremediation due to microbial shift.
[Reply]: Thank you for highlighting this critical aspect.The observed microbial shifts under FXP and MFZ exposure have critical implications for soil-plant systems: (1) Plant growth. ① Positive: Enriched stress-tolerant endophytes (Azorhizobium, Luteibacter) may enhance plant resilience to abiotic stress via phytohormone production. Upregulated Lysobacter (rhizosphere) suppresses soil-borne pathogens, indirectly supporting root health. ② Negative: Suppression of Bradyrhizobium (a nitrogen-fixing genus) could reduce nitrogen availability, potentially stunting plant growth in nitrogen-limited soils. (2) Nutrient cycling.① Carbon & organic matter: Reduced oligotrophic taxa (RBG-16-49-21) may slow organic matter decomposition, limiting nutrient mineralization.② Nitrogen: Transient decline in nitrogen fixers (Bradyrhizobium) and nitrifiers (Nitrospira) could disrupt N-cycling, necessitating fertilizer supplementation.③ Sulfur: Enriched Thiobacillus (sulfur oxidizer) may enhance sulfur availability, aiding pesticide detoxification. (3) Bioremediation. Enhanced degradation: Enrichment of degraders (Sphingomonas) suggests accelerated pesticide breakdown, reducing environmental persistence. These findings have been integrated into the revised manuscript (Lines 304-320).
Question 7: The manuscript should be thoroughly revised by an English language expert to ensure clarity, grammatical accuracy, and professional tone.
[Reply]: We sincerely appreciate the reviewer’s valuable feedback on improving the language quality of our manuscript. We confirm that the revised manuscript has undergone multi-stage linguistic refinement to meet the highest standards of scientific English. All changes are highlighted in the tracked-changes version for transparent evaluation.
Question 8: Ensure that all scientific names are italicized.
[Reply]: Thanks for your notice. We have revised it.

Reviewer 2 Report
Comments and Suggestions for Authors
The authors presented an interesting study on the effects of fluxapyroxad and mefentrifluconazole application on the microbiota and metabolome associated with rice cultivation. The data obtained are valuable, but they do not emphasize the most outstanding results and fall into the descriptive ones. There are also many methodological aspects that prevent to know the strategy of analysis and the reproducibility of your study, we invite you to follow the following recommendations.
>Title
The title is general, mention the fungicides under study.
>Introduction
Please update with more information on recent work from 2025, including links.
Add information on the mechanisms of action of fungicides and their classification.
Information on the main rice pathogens and which of them are resistant to fungicides.
Add information on the importance of studying microbiota in the context of fungicide use and the potential use of microorganisms as indicators of soil and plant health.
>Materials and methods
Be sure to write numbers with letters at the beginning of a sentence, e.g. L-106 “400 g/L”.
Describe DNA extraction methodology and sample quantities used.
How did you ensure that endophytic and non-epiphytic microbiota were analyzed?
Describe in more detail the generation of the sequencing libraries
Indicate the sequencing depth and the size of the reads obtained.
Make a specific and detailed section of the bioinformatics analysis with the versions of the software used.
Include:
- Criteria for the selection of good quality reads (Q phred).
- Sequence joining
-Final size of the amplicon used for classification
- Database for taxonomy
- Filtering strategy to retain only bacterial 16S rRNA reads and not host 16S rRNA reads.
- Calculation of alpha and beta diversity indices (NMDS, PcoA).
- Differential abundance analysis (LEfse, DESEq)
- Statistical tests between treatments (PERMANOVA, ANOSIM)
- Venn diagrams to identify core microbiota.
- Functional levels analyzed with PICRUSt2
- Bioproject ID where the raw sequences were deposited.
It is necessary to detail the strategy of metabolomic analysis especially when dealing with non-targeted analysis, where there are many unknown metabolites.
Please, add software for analysis of raw spectra, determination of charge mass ratio, differential abundance analysis, chemical classification of metabolites (against database or structural prediction).
Include correlation analysis to associate microbiota abundance data with metabolite abundance, for example, a network or CCA analysis.
>Results and discussion
The recommendation is to separate the results from the discussion. Especially in omics analyses, such as the microbiome and metabolome, many results are obtained. Consequently, this section is often further overloaded with the addition of the discussion.
In the results, detail the most important findings and support the information with supplementary material to avoid unnecessary repetition.
In the discussion, focus on describing the explanation of what was found among the multiple analyses and the relevance of the association between microbiota and metabolome. The implications of the above in rice cultivation.
>Conclusion
Take up the modifications made to the content of the manuscript and add some lines on future lines of research.
Author Response
Response to Reviewer 2: The authors presented an interesting study on the effects of fluxapyroxad and mefentrifluconazole application on the microbiota and metabolome associated with rice cultivation. The data obtained are valuable, but they do not emphasize the most outstanding results and fall into the descriptive ones. There are also many methodological aspects that prevent to know the strategy of analysis and the reproducibility of your study, we invite you to follow the following recommendations.
Some comments are as follows:
Question 1: >Title
The title is general, mention the fungicides under study.
[Reply]: Thanks for your advice. We have revised it.
Question 2: >Introduction
Please update with more information on recent work from 2025, including links.
Add information on the mechanisms of action of fungicides and their classification.
Information on the main rice pathogens and which of them are resistant to fungicides.
Add information on the importance of studying microbiota in the context of fungicide use and the potential use of microorganisms as indicators of soil and plant health.
[Reply]: Thank you for your guidance. All requested updates (recent studies in 2025, fungicide mechanisms/classification, rice pathogen resistance and microbial roles) have been integrated into the Introduction.
Question 3: >Materials and methods
Be sure to write numbers with letters at the beginning of a sentence, e.g. L-106 “400 g/L”.
Describe DNA extraction methodology and sample quantities used.
How did you ensure that endophytic and non-epiphytic microbiota were analyzed?
Describe in more detail the generation of the sequencing libraries
Indicate the sequencing depth and the size of the reads obtained.
Make a specific and detailed section of the bioinformatics analysis with the versions of the software used.
Include:
- Criteria for the selection of good quality reads (Q phred).
- Sequence joining
-Final size of the amplicon used for classification
- Database for taxonomy
- Filtering strategy to retain only bacterial 16S rRNA reads and not host 16S rRNA reads.
- Calculation of alpha and beta diversity indices (NMDS, PcoA).
- Differential abundance analysis (LEfse, DESEq)
- Statistical tests between treatments (PERMANOVA, ANOSIM)
- Venn diagrams to identify core microbiota.
- Functional levels analyzed with PICRUSt2
- Bioproject ID where the raw sequences were deposited.
It is necessary to detail the strategy of metabolomic analysis especially when dealing with non-targeted analysis, where there are many unknown metabolites.
Please, add software for analysis of raw spectra, determination of charge mass ratio, differential abundance analysis, chemical classification of metabolites (against database or structural prediction).
Include correlation analysis to associate microbiota abundance data with metabolite abundance, for example, a network or CCA analysis.
[Reply]: Thank you for your guidance. (1) Numerical formatting: all numerical values at sentence beginnings are written as words. (2) DNA extraction methodology and sample quantities: Genomic DNA was extracted from 24 soil samples (6 replicates × 4 treatments: CK7D, LF7D, CK21D and LF21D) using the CTAB protocol: Homogenization in CTAB buffer (2% CTAB, 1.4 M NaCl). Chloroform-isoamyl alcohol purification. Ethanol precipitation and resuspension in TE buffer. The specific DNA extraction steps were added to the Supporting information. (3) Endophytic and non-epiphytic microbiota: Surface sterilization: roots were treated with 3% NaClO (5 min) and 70% ethanol (2 min) to remove epiphytes. Validation: post-sterilization, surface rinse solutions were subjected to PCR amplification (universal 16S rRNA primers) to confirm the absence of residual microbial DNA (negative controls). Endophytes: DNA was extracted from surface-sterilized root tissues, ensuring only internally colonized microbes were analyzed. Rhizosphere microbiota (non-epiphytic microbiota): DNA was extracted from rhizosphere soil (adhering to roots but not sterilized), representing microbes on root surfaces and surrounding soil. Negative controls: sterilized blank root samples were processed to validate sterilization efficacy. Positive controls: DNA from unsterilized roots confirmed the presence of epiphytes. This methodology effectively isolates endophytic (intra-root) and rhizosphere-associated (non-epiphytic) communities. Detailed protocols are provided in Supporting information. (4) The steps were ① Primers: V3-V4 region amplified using 341F/805R. ② PCR: Purified with AMPure XT beads, quantified via Qubit. ③ Sequencing: Illumina NovaSeq (paired-end 250 bp) at LC-Bio (Hangzhou). The information was added to the Sequencing library generation section of the Supporting information. (5) Sequencing depth and read size. Depth: 65258-84282 high-quality reads per sample (Line 266). Read length: 250 bp paired-end. The information was added to the Sequencing library generation section of the Supporting information. (6) Bioinformatics analysis (detailed). Software versions: DADA2 (v1.26.0) for ASV clustering (100% similarity). SILVA v138 and NT-16S databases for taxonomy. PICRUSt2 (v2.5.2) for functional prediction (KEGG). Quality Control: reads filtered at Q ≥ 20 (Phred score). Host DNA removed by excluding eukaryotic sequences during SILVA classification. Amplicon size: ~460 bp (V3-V4). Diversity indices: Alpha: Chao1, Shannon, Simpson (calculated in QIIME2). Beta: PCoA (Bray-Curtis) and NMDS (STAMP). Differential abundance: LEfSe (LDA > 2.0), DESeq2 (FDR < 0.05). Statistics: PERMANOVA/ANOSIM (PRIMER7) for treatment effects. Core microbiota: The core microbiota and differential microbiota were described through bar charts in Figure 3 and Supporting information (Figs. S2 and S3). Data deposition: The process of submitting sequencing data to the NCBI is currently in progress. (7) Metabolomic Analysis. Non-targeted workflow: LC-MS: Vanquish UHPLC/Q-Exactive MS (Thermo Fisher Scientific, USA). Software: Compound Discoverer 3.1.0 (Thermo Fisher Scientific, USA) for raw spectra, determination of charge mass ratio and compound identification. Metabolite identification was combined with the in-house metabolite secondary Mass spectrometry library, mzCloud, mzVault, Mass list, OTCML, ChemSpider (KEGG, LipidMaps, PlantCyc, Planta Piloto de Quimica), etc. Finally, information such as the molecular weight, retention time, peak area and metabolite identification results of each ion was obtained. The detailed informations of the analysis software involved in the metabolome were added to the Supporting information. Annotation: HMDB and KEGG databases. Structural Prediction: SIRIUS (v5.5.6). Differential analysis: 1) ratio >= 2 或者 ratio <= 1/2;2) p value <= 0.05;3) VIP ≥ 1. Correlation: We appreciate the suggestion to perform CCA analysis. While our current study utilized correlation heatmaps to visualize pairwise associations between microbial taxa and differential expression of metabolites (Fig. S9), we acknowledge that CCA could further explore multivariate relationships with environmental variables. However, given the high dimensionality and complexity of our dataset, correlation heatmaps were prioritized. We will incorporate CCA in future work.
Question 4: >Results and discussion
The recommendation is to separate the results from the discussion. Especially in omics analyses, such as the microbiome and metabolome, many results are obtained. Consequently, this section is often further overloaded with the addition of the discussion.
In the results, detail the most important findings and support the information with supplementary material to avoid unnecessary repetition.
In the discussion, focus on describing the explanation of what was found among the multiple analyses and the relevance of the association between microbiota and metabolome. The implications of the above in rice cultivation.
[Reply]: Thank you for the constructive feedback. We have restructured this chapter to clearly demarcate results (data presentation) from discussion (interpretation and implications). Below are the key revisions: (1) Use subheadings: Separate the results and discussion content through subheadings (such as "3.6.1 Metabolomic profiling", "3.6.2 Impact on rice aroma-related phenolic compounds", "3.6.3 Metabolic pathway regulation") to make the structure clearer. (2) In the results, Metabolomic profiling: Key findings (e.g., 3,512 metabolites identified, DEMs categorized into lipid or benzenoid classes) are presented concisely, supported by Supplementary Tables/Figures (Fig. S5-S9); Detailed metabolite classifications and pathway enrichments added to Supplementary Tables S6-S7 (3) Focus for discussion section revisions, the revised discussion emphasizes the mechanistic links between microbiota shifts and metabolome reprogramming, and their practical implications for rice cultivation. Link enriched microbial taxa (such as Azorhizobium) to metabolic pathways (such as TCA cycle acceleration, phenylpropanoid suppression) under pesticide stress. Enriched stress-tolerant endophytes (e.g., Azorhizobium) could be harnessed as bioinoculants to enhance resilience, reducing pesticide dependency.
Question 5: >Conclusion
Take up the modifications made to the content of the manuscript and add some lines on future lines of research.
[Reply]: Thank you for your guidance. We have revised it in Lines 476-490: This study systematically validated a analytical method for quantifying FXP and MFZ residues, achieving compliance with EU guidelines. Dissipation dynamics of both fungicides revealed matrix-dependent persistence, with MFZ exhibiting prolonged half-lives in brown rice (T1/2 = 57.76 d) and soil (T1/2 = 25.67 d), highlighting its potential environmental and dietary risks. While final residues in brown rice (0.031 μg/g FXP; 0.011 μg/g MFZ) fell below regulatory limits (China: 1 mg/kg; CAC: 5 mg/kg), transient residue spikes in rice husk (14 d) and soil-microbe-mediated degradation patterns underscore the need for post-application monitoring. Microbial analysis demonstrated rhizosphere resilience, with alpha diversity recovering by 21 d post-treatment, whereas endophytes exhibited stability linked to plant metabolite buffering. Root exudates enriched stress-adapted taxa, suggesting microbial modulation of pesticide impacts. These findings advocate for optimized application timing aligned with microbial recovery phases and targeted bioaugmentation strategies to mitigate leaching risks. Future studies should prioritize field trials to validate long-term soil health impacts under real-world agricultural practices.

Round 2
Reviewer 2 Report
Comments and Suggestions for Authors
The authors substantially improved the resubmitted manuscript. The introduction was updated, and the methods are now more descriptive and reproducible. The results are now more clear. However, the section on bioinformatics and metabolomics analysis must be included in the main manuscript. I also considered including a figure showing functional prediction analysis. When information is not provided in the text, please indicate that it is found in the supplemental material.
For example, the CTAB extraction mentions that it was extracted following the manufacturer's description; however, a homemade method was actually used, for which there are no references. Check for similar cases.
Author Response
We sincerely appreciate your constructive feedback. Below are our point-by-point responses:
Response to Reviewer 2:
The authors substantially improved the resubmitted manuscript. The introduction was updated, and the methods are now more descriptive and reproducible. The results are now more clear.
Question 1: However, the section on bioinformatics and metabolomics analysis must be included in the main manuscript.
[Reply]: Thanks for your advice. We have now integrated detailed descriptions of bioinformatics (16S rRNA sequencing) and metabolomics (UPLC-MS/MS data processing) into Sections 2.4 and 2.5 of the main manuscript.
Question 2: I also considered including a figure showing functional prediction analysis. When information is not provided in the text, please indicate that it is found in the supplemental material.
[Reply]: As suggested, the figure showing functional prediction analysis (Fig. S5) was provided in the supplemental material.
Question 3: For example, the CTAB extraction mentions that it was extracted following the manufacturer's description; however, a homemade method was actually used, for which there are no references.
[Reply]: We apologize for the ambiguity. The extraction method of CTAB-based DNA was checked and revised strictly following the manufacturer’s protocol, as now explicitly stated in Supporting information section 1.1 (includes key protocol steps: e.g., lysis with CTAB buffer, chloroform:isoamyl alcohol purification, isopropanol precipitation). No homemade modifications were applied.
Question 4: Check for similar cases.
[Reply]: Thank you again for your meticulous review. We thoroughly reviewed all experimental protocols in the manuscript to ensure consistency between described methods and their citations.
